# Mitochondrial Ceramide Effects on the Retinal Pigment Epithelium in Diabetes

**DOI:** 10.3390/ijms21113830

**Published:** 2020-05-28

**Authors:** Yan Levitsky, Sandra S. Hammer, Kiera P. Fisher, Chao Huang, Travan L. Gentles, David J. Pegouske, Caimin Xi, Todd A. Lydic, Julia V. Busik, Denis A. Proshlyakov

**Affiliations:** 1Department of Physiology, Michigan State University, East Lansing, MI 48824, USA; levitsk2@msu.edu (Y.L.); shammer@msu.edu (S.S.H.); fishe247@msu.edu (K.P.F.); chaohuang@uabmc.edu (C.H.); gentlest@msu.edu (T.L.G.); lydictod@msu.edu (T.A.L.); 2Department of Chemistry, Michigan State University, East Lansing, MI 48824, USA; pegousk1@msu.edu (D.J.P.); caiminxi@gmail.com (C.X.)

**Keywords:** diabetes, retinopathy, acid sphingomyelinase, mitochondria, ceramide, sphingolipid, respirometry, metabolism, dyslipidemia

## Abstract

Mitochondrial damage in the cells comprising inner (retinal endothelial cells) and outer (retinal pigment epithelium (RPE)) blood–retinal barriers (BRB) is known to precede the initial BRB breakdown and further histopathological abnormalities in diabetic retinopathy (DR). We previously demonstrated that activation of acid sphingomyelinase (ASM) is an important early event in the pathogenesis of DR, and recent studies have demonstrated that there is an intricate connection between ceramide and mitochondrial function. This study aimed to determine the role of ASM-dependent mitochondrial ceramide accumulation in diabetes-induced RPE cell damage. Mitochondria isolated from streptozotocin (STZ)-induced diabetic rat retinas (7 weeks duration) showed a 1.64 ± 0.29-fold increase in the ceramide-to-sphingomyelin ratio compared to controls. Conversely, the ceramide-to-sphingomyelin ratio was decreased in the mitochondria isolated from ASM-knockout mouse retinas compared to wild-type littermates, confirming the role of ASM in mitochondrial ceramide production. Cellular ceramide was elevated 2.67 ± 1.07-fold in RPE cells derived from diabetic donors compared to control donors, and these changes correlated with increased gene expression of *IL-1β*, *IL-6*, and *ASM*. Treatment of RPE cells derived from control donors with high glucose resulted in elevated *ASM*, vascular endothelial growth factor (*VEGF*), and intercellular adhesion molecule 1 (*ICAM-1*) mRNA. RPE from diabetic donors showed fragmented mitochondria and a 2.68 ± 0.66-fold decreased respiratory control ratio (RCR). Treatment of immortalized cell in vision research (ARPE-19) cells with high glucose resulted in a 25% ± 1.6% decrease in citrate synthase activity at 72 h. Inhibition of ASM with desipramine (15 μM, 1 h daily) abolished the decreases in metabolic functional parameters. Our results are consistent with diabetes-induced increase in mitochondrial ceramide through an ASM-dependent pathway leading to impaired mitochondrial function in the RPE cells of the retina.

## 1. Introduction

Diabetic retinopathy is the leading cause of blindness among working-age adults, representing a large socioeconomic burden on society. To date, medical and surgical treatment options have been revolutionary; however, indications for treatment rely on advanced markers of disease. Development of effective treatments for early stages of the disease require elucidation of the underlying biochemical pathophysiology.

The retina is composed of a highly ordered and bioenergetically active neural tissue, perfused by two independent vasculatures. The retinal and choroidal vessels, supplying the inner and outer retina, respectively, regulate molecular exchange across the inner and outer blood–retinal barriers. Breakdown of these barriers results in clinically observable lesions, such as microaneurysms and hemorrhages, ultimately leading to retinal hypoxia or ischemia and disease progression [1]. The inner blood–retinal barrier, consisting of non-fenestrated retinal endothelial cells and pericytes, has been the focus of many studies, but the outer barrier has received comparatively less attention [1,2,3]. The choriocapillaris, a vascular layer supplying circulation to the outer third of the retina, consists of a fenestrated endothelium separated by Bruch’s membrane from the retinal pigment epithelium (RPE). The RPE provides a barrier function, with expression of tight junction proteins and regulation of transcellular water, ion, and metabolite transport by polarized expression of transporters [4,5]. Apart from regulating the osmotic and ionic balance of the outer retina, the RPE plays a key role in vision by phagocytosing shed photoreceptor outer segments and recycling retinoids for the visual cycle [4,5]. Therefore, RPE dysfunction can contribute to the hypoxic conditions common in DR, and to the fluid and ion fluxes thought to cause diabetic macular edema [6,7,8].

Diabetic retinopathy is a neurovascular complication of diabetes resulting from chronic exposure to hyperglycemia, dyslipidemia, and inflammation. Diabetic dyslipidemia leads to changes in systemic and local lipid metabolism that drive the pro-inflammatory and pro-apoptotic cellular changes typical of diabetic retinopathy (DR) [9]. Ratios of key sphingolipid species, such as ceramide and sphingosine-1-phosphate, are a major factor in sphingolipid metabolism and play key roles in cell fate [10]. These ratios are termed a “sphingolipid rheostat” due to their importance in determining cell growth, proliferation, and apoptosis. In particular, ceramides are bioactive sphingolipid species which regulate cell stress responses [11,12]. Ceramides can be synthesized de novo from serine and palmitate or salvaged from other sphingolipid species, depending on the physiological state of the tissue [13]. Structurally, ceramide is composed of a sphingoid base and an exchangeable fatty acid. The chain length of the exchangeable fatty acid determines the biological effect of the ceramide. While short-chain ceramides (<20 carbons) are associated with pro-apoptotic effects, long-chain ceramides (>20 carbons) exert a protective effect on cells [14]. Ceramide is produced from sphingomyelin by hydrolysis of the phosphocholine head group, and the enzymes which catalyze this reaction, the sphingomyelinases, are distinguished by the pH at which they show optimum activity [13].

Acid sphingomyelinase (ASM) catalyzes sphingomyelin hydrolysis in lysosomes and at the plasma membrane, showing relative specificity for producing short-chain ceramides [15,16]. The ASM-knockout mouse has been well characterized as an animal model of Neimann Pick disease, showing remarkable resistance to cellular toxicity stemming from a variety of stressors such as hypoxia, radiation, and ischemia-reperfusion injury [12,15,17]. Specifically, the ASM-knockout mouse is resistant to retinal ischemia-reperfusion injury, confirming the central role of ceramide generation in the response to cell stress [15]. Studies in animal and cell culture models of DR have shown that it is the ASM, rather than the neutral SM, that is increased in the retina and retinal cells [15]. Moreover, inhibition of ceramide synthase, the central enzyme of the de novo ceramide production pathway, had no effect on cytokine-induced pro-inflammatory changes in the retina and retinal cells [3], further supporting the central role of ASM in ceramide-mediated retinal pathology.

Reports of direct effects of ceramide on mitochondrial structure and function [18,19,20,21,22,23] prompted us to consider whether diabetes-induced ASM upregulation might lead to mitochondrial ceramide accumulation and, in turn, to structural and functional changes. Overall changes in sphingolipid levels have been documented in the diabetic retina, but elevated ceramide levels were not evident. Instead, decreases in ceramide species were compensated with increases in hexosylceramides, consistent with an increase in ceramide glycosylation in diabetes [24]. In the current study, we examine diabetes-induced changes in retinal mitochondria-specific ceramide, and demonstrated that changes in mitochondrial structure and function occur in an ASM-dependent manner, in contrast to the sphingolipid changes in the whole retina.

## 2. Results

### 2.1. Diabetes Results in Retinal Mitochondrial Ceramide Accumulation

We used an streptozotocin (STZ)-induced diabetic rat model to determine whether upregulation of ASM expression and activity in cells comprising the inner and outer blood–retinal barriers (BRBs) [15] contributed to the mitochondrial ceramide accumulation that further leads to cell damage [25]. Mitochondria were isolated from control and diabetic rat retinas using differential centrifugation protocols, followed by lipid extraction using chloroform, methanol, and water [26] and Orbitrap high-resolution/accurate mass mass spectrometry (MS) and MS/MS analysis. Samples were normalized based on total mitochondrial protein, and sphingolipid peaks were compared to synthetic sphingolipid internal standards incorporated in each run. As presented in Figure 1A,B, diabetes resulted in a decrease in endogenous levels of mitochondrial ceramide and sphingomyelin, consistently with previous reports [24]. To quantify changes in sphingolipid composition, total detected sphingolipid abundances were summed, and sphingolipid species were expressed as a percentage of total sphingolipids. This approach revealed significant increases in relative ceramide levels and decreases in the relative sphingomyelin levels in retinal mitochondria isolated from STZ-induced diabetic rat retinas (7-week duration) compared to controls (Figure 1C, left). To test the role of ASM in the control of mitochondrial ceramide more directly, sphingolipid profiles of mitochondria prepared from ASM-knockout mice were similarly analyzed. In contrast to the diabetes-induced increase in the ceramide-to-sphingomyelin ratio, depletion of ASM resulted in lower relative levels of ceramide versus sphingomyelin compared to wild-type controls (Figure 1C, right), confirming that ASM plays an important role in mitochondrial sphingolipid dynamics. 

### 2.2. Diabetes Results in Pro-Inflammatory Changes in Human Retinal Pigment Epithelium (RPE) Cells

Whole-retina preparations, as shown in Figure 1, lack the RPE layer, a site of diabetes-induced ASM upregulation [15]. We therefore sought to determine separately whether RPE cells demonstrated similar diabetes-induced changes.

Results of fluorescent ceramide staining in control- and diabetic-derived cultured human RPE cells are presented in Figure 2A, and demonstrated an average 2.7-fold increase in cellular ceramide staining of diabetic-derived RPE cells compared to controls (Figure 2B). Analysis of inflammatory gene expression in the same cells showed significant increases in *IL-1β* and *IL-6*, and a trend toward increased ASM expression in diabetic-derived RPE cells compared to controls (Figure 2C), which was consistent with the increases in ceramide observed by immunohistochemistry. Furthermore, in vitro treatment of control-derived RPE cells with 25 mM glucose for 72 h led to significant increases in ASM, VEGF, and ICAM1 mRNA compared to untreated control-derived RPE cells (Figure 2D), supporting their roles in hyperglycemic response.

### 2.3. Diabetes Results in Mitochondrial Fragmentation in Human RPE Cells

As mitochondria are known to accumulate ceramide [27], and we demonstrated that diabetes changes the ceramide-to-sphingomyelin ratio in retinal mitochondria (Figure 1), we next sought to determine whether structural and functional changes could be detected in mitochondria isolated from control- or diabetic-derived RPE cells. Figure 3A demonstrates staining with MitoTracker Green, used to reveal the expected reticular mitochondrial network in the control RPE cells. This network appeared to be disrupted in the diabetic-derived RPE cells, which had predominantly round and fragmented mitochondria. These changes are further presented in Appendix A as an animated 3D reconstruction. Quantitation of morphological features revealed that the average mitochondrial length in diabetic-derived RPE cells was 1.2 ± 0.57 μm (*n* = 3), whereas control-derived RPE cells’ mitochondria were 3.4 ± 0.78 μm (*n* = 3) in length, (Figure 3B).

### 2.4. Diabetes Induces Acid Sphingomyelinase (ASM)-Mediated Changes in Mitochondrial Function of Human RPE Cells

To determine whether the structural changes of mitochondria were correlated with detectable functional differences, we used microrespirometry to examine oxidative phosphorylation in control- and diabetic-derived RPE cells [28]. In this approach, the flow of oxygenated medium over adherent cells is intermittently stopped and respiration leads to a steady consumption of oxygen, seen as periodic downward slopes in the O_2_ concentration traces (Figure 4A). Ensuing resumption of flow re-oxygenates the sample and the measurement is repeated. Following growth to confluency, RPE cells were transferred into the microrespirometer and perfused with a medium containing glucose, lactate, and pyruvate as substrates, supplemented with the ATP-synthase inhibitor oligomycin (Figure 4A, leak). Respiratory activity in this state is limited by a high proton motive force and predominantly represents proton leakage through the inner mitochondrial membrane [29]. No substantial differences between sample groups were observed in this state, suggesting a lack of diabetes-induced changes to inner mitochondrial membrane proton permeability. Next, the maximal respiratory rate was assessed by dissipation of the proton motive force with the chemical uncoupler, carbonyl cyanide m-chlorophenylhydrazone (CCCP, uncoupled). In this state, control over respiration is shifted to substrate delivery pathways and innate turnover capacity of the electron transport chain. Dissipation of the proton motive force with CCCP resulted in an increase in oxygen consumption rates of RPE cells over that observed with oligomycin alone. While the expected increase in respiration due to uncoupling was observed in control RPE cells and in diabetic cells with desipramine pretreatment, little response to the uncoupler was observed in the diabetic cells without desipramine pretreatment. The latter observation indicated that mitochondria in resting diabetic RPE cells operate close to the maximal respiratory activity, which is limited by electron transport chain turnover or substrate delivery. Subsequent perfusion with potassium cyanide resulted in complete inhibition of mitochondria-dependent oxygen consumption, evident in all experimental groups and used to correct for non-mitochondrial oxygen-consuming processes. The relative changes in the oxygen consumption between three conditions were used to calculate a respiratory control ratio (RCR, Equation (1)), a quantitative measure of mitochondrial fitness [30].
(1)$$Respiratory\;Control\;Ratio\;\left(RCR\right)=\frac{{\left(Oligomycin+CCCP\right)}_{OCR}-{\left(KCN\right)}_{OCR}}{{\left(Oligomycin\right)}_{OCR}-{\left(KCN\right)}_{OCR}}$$

As shown in Figure 4B, diabetic-derived RPE cells displayed a significantly decreased RCR compared to control-derived RPE cells (1.41 ± 0.27 vs. 3.78 ± 0.59). This difference was abolished by perfusion of diabetic-derived RPE cells with 15 µM desipramine, an ASM inhibitor, [31,32] for 1 h, which increased the RCR to 5.00 ± 1.78.

### 2.5. Mitochondrial ASM Contributes to Impaired Mitochondrial Function In Vitro

Accumulation of ceramide at the expense of sphingomyelin with concomitant change in ceramide/sphingomyelin ratio (vide supra) suggests that it is the result of sphingomyelin hydrolysis, a reaction catalyzed by ASM. This raises the question of whether ceramide accumulation in mitochondria is due to the activity of mitochondrial ASM or to the transport of ceramide from remote sites. The presence of intrinsic ASM in mitochondrial membranes is demonstrated in Figure 5. In addition to mitochondria, ASM is known to be present in the lysosomes and the plasma membrane. Although a high degree of separation between the mitochondrial and plasma membrane fractions is easily achievable by standard methodology, the lysosomes and mitochondria and are much harder to separate due to very similar size, shape, and density characteristics [33,34,35,36]. To conclusively localize ASM to mitochondrial membranes, we obtained mitochondrial preparations with increasing levels of purity from human RPE cells and subjected them to immunoblotting for (i) ASM; (ii) the mitochondrial outer membrane marker voltage-dependent anion channel (VDAC); and (iii) the lysosomal membrane marker lysosomal-associated membrane protein 1 (LAMP-1). As shown in Figure 5A and Appendix A, mitochondria prepared by standard differential centrifugation protocols showed the presence of VDAC and LAMP-1 (crude), indicating co-purification of lysosomes in mitochondrial preparations. However, further purification by successive centrifugations at 8000× *g* (pure) resulted in significant depletion of the lysosomal marker LAMP-1 relative to the mitochondrial marker VDAC. Each fraction was sampled at varying concentrations, ensuring fidelity of optical density quantitation, which revealed that extra purification yielded a two-fold depletion of LAMP-1 (Figure 5B, left panel). The same samples were then probed for the level of ASM, which is known to localize to the lysosome. If most of the ASM shown in Figure 5 originated from the lysosomal compartment, the ratio of ASM to VDAC would follow that of LAMP-1. The experimental results demonstrated only a small ASM depletion in the purified mitochondrial sample, and optical density quantitation revealed an apparent enrichment of ASM when normalized to VDAC density (Figure 5B, right panel). 

These results were replicated in mitochondria isolated from ARPE-19 using a similar protocol (Figure 5C and Appendix A). In this case, the mitochondrial fractions were examined by immunoblotting after initial differential centrifugation (D.C.) and again after a second sucrose-step density ultracentrifugation (U.C.). Consistent with the results from the human RPE cells (Figure 5B), the ultracentrifugation resulted in a two-fold depletion of LAMP-1 normalized to VDAC (Figure 5D, left panel) compared to the differential centrifugation preparation of mitochondria. ASM enrichment was also observed after ultracentrifugation when normalized to VDAC in ARPE-19 samples (Figure 5D, right panel).

To examine the consequences of increased mitochondrial ASM expression and the resulting mitochondrial ceramide accumulation, we assessed changes in citrate synthase activity in response to high glucose treatment of ARPE-19 (Figure 6). Citrate synthase resides exclusively in the mitochondrial matrix and catalyzes the condensation of acetyl-CoA and oxaloacetate to citrate, in the first step of the tricarboxylic acid cycle. As such, it is widely used as a mitochondrial content marker [37]. 

The effect of glucose concentration on citrate synthase activity in ARPE-19 cells is presented in Figure 6. No statistically significant effect of high glucose was observed after 24 h (*n* = 5, *p* > 0.05). At 48 h, the 25 mM glucose treatment increased citrate synthase activity to 164.3% ± 5.8% vs. 5.5 mM glucose control, followed by the reduction in activity to 75% ± 1.6% of the control at 72 h (*p* < 0.05, *n* = 5). To evaluate whether these glucose-induced changes in citrate synthase activity was mediated by ASM, parallel measurements were conducted on ARPE-19 cells incubated in high-glucose conditions with daily, intermittent treatments with 15 µM desipramine. These treatments abolished the biphasic hyperglycemia-induced response in citrate synthase activity, and the desipramine-treated group displayed a time profile closely corresponding to that of the control group (*p* > 0.05, *n* = 5).

## 3. Discussion

Diabetes is a multifactorial pathological process resulting in micro- and macrovascular complications. Diabetic retinopathy is a common microvascular complication of diabetes, which results from hyperglycemia, dyslipidemia, and chronic inflammatory changes in the retina leading to blood–retinal barrier breakdown and disease progression [38]. Diabetic dyslipidemia results in both systemic and local changes to lipid metabolism and, in the retina, contributes to the pro-apoptotic changes seen in the inner and outer blood–retinal barrier cellular components [9]. 

We previously demonstrated that diabetes leads to enhanced ASM expression predominantly in retinal endothelial- and retinal pigment epithelial cells [15] suggesting that sphingomyelin hydrolysis is the primary cause of cellular ceramide accumulation. Despite these findings, measurements of sphingolipid composition in diabetic rodent retinas revealed that ceramide levels are, in fact, decreased whereas glucosylceramides are increased [24]. Such results suggest hyperglycemia-induced diversion of ceramide toward the glycosylated forms in total retinal sphingolipid pools. The increased glycosylation in the diabetic retina was attributed to increases in uridine diphosphate glucose (UDP-glucose) production through the pentose pathway, rather than changes in enzymatic activity. As the pentose phosphate pathway occurs in the cytoplasm, we argue that an increase in glucosylceramide production due to higher UDP-glucose availability would be limited to the cytoplasm, rather than the mitochondria. In contrast to whole-retina sphingolipid measurements, we show in Figure 1A that diabetes-induced increases in relative ceramide levels can be detected in mitochondria after subcellular fractionation of whole retina. Similarly, mitochondria isolated from the retinas of ASM-knockout animals displayed an inversion of the ceramide-to-sphingomyelin ratio (Figure 1B), demonstrating the direct connection between ASM activity and mitochondrial ceramide accumulation. It is worth mentioning that the degree of these changes in the barrier cells was likely underestimated because the major component of the mitochondrial preparations from whole retina originate from photoreceptors. Diabetes-induced increase in ASM expression and activity is the highest in the cells that make up the BRB, namely REC and RPE cells [3,15,16]. Smaller changes are observed in the Muller cells and microglia, and no changes are seen in the photoreceptors [15]. The changes in the whole-retina mitochondria preparations were thus diluted by the large population of the non-changing photoreceptor mitochondria and by the mitochondrial ceramide from endothelial, Muller, and microglia cells, which shows smaller changes.

Here, we focused on the role of ASM-dependent sphingomyelin hydrolysis in barrier cells. ASM-dependent mitochondrial ceramide accumulation is strongly supported by our present finding that a population of cellular ASM can be localized to mitochondrial membranes in RPE cells (Figure 5). These results are consistent with our previous reports that diabetes-induced ASM upregulation is a key player in blood–retinal barrier breakdown, and provide evidence for a proposed mechanism of metabolic dysfunction in retinal cells mediated by the accumulation of cellular ceramide. Although our results support the role of mitochondrial ASM in the observed changes, we cannot presently rule out the contributions of alternative pathways, such as neutral-sphingomyelinase- and/or reverse-ceramidase-mediated mitochondrial ceramide generation, as described in other systems [25,27]. Our previous data show that neutral sphingomyelinase expression does not change in the diabetic retina [15], and that inhibition of ceramide synthase has no effect on cytokine-induced pro-inflammatory changes in the retina and retinal cells [3]. In combination with the sensitivity of both the diabetes-induced changes in RCR and citrate synthase to desipramine reported here, we strongly argue that ASM plays a key role in RPE cell mitochondrial dysfunction. Alternatively, stress-induced production and transport of ceramide to mitochondria from distal sites has also been reported [39].

In this work we focused on the RPE cells, a cellular component of the outer blood–retinal barrier. As RPE cells were not a part of the whole-retina preparations, we examined mitochondria from the control and diabetic donors RPE cells separately from the whole-retina mitochondria. Human RPE cell culture could be used due to a well-known metabolic memory legacy effect. Metabolic memory was first described in diabetic patients as a prolonged effect of early glycemic control on the development of diabetic complications, even after glycemic control is established later in the course of disease progression [40]. The metabolic memory phenomenon is well accepted in the field of diabetic complications [40,41,42,43]. The molecular mechanisms underpinning sustained metabolic memory are not fully understood. Recent work, however, has demonstrated that epigenetic modifications to mtDNA mismatch-repair machinery result in decreased transcript levels, decreased mitochondrial localization, and accumulation of mtDNA mutations [38]. As mtDNA is particularly vulnerable to ROS-induced DNA mutations, decreased functioning of repair machinery results in accumulation of damaged oxidative phosphorylation complexes and, ultimately, impairment of oxidative phosphorylation as a whole [39]. As these changes accumulate over time, they perpetuate a vicious cycle of oxidative stress and sustained inflammatory changes, leading to the progression of diabetic complications despite correction of diabetic hyperglycemia.

These effects have been shown to occur in animal models as well as in cell culture models. Cells isolated from diabetic donor retinas or animal models retain their diabetic metabolic phenotype for several passages [40,41,42,43,44,45]. Human control and diabetic donor cells were previously shown to display metabolic memory characteristics right after the isolation and for up to eight passages [40]. Dysfunction of these cells is implicated in the development of diabetic macular edema and they represent a site of significant diabetes-induced ASM upregulation [6,15]. Despite culturing control- and diabetic-derived RPE cells under identical, euglycemic, conditions, we detected increased ceramide and inflammatory gene expression in diabetic-derived RPE cells compared to controls (Figure 2). Furthermore, we showed that control-derived RPE cells retained their sensitivity to the diabetic milieu, as high-glucose treatment resulted in enhanced inflammatory gene expression (Figure 2D). Accompanying these changes, diabetic-derived RPE cells displayed fragmented mitochondria and impaired mitochondrial-dependent metabolism (Figure 3 and Figure 4). These results support the metabolic memory hypothesis, implying that diabetes induces permanent changes to cellular metabolism in the long term, despite achievement of a euglycemic state.

Our results are consistent with previous reports detailing diabetes-induced mitochondrial fragmentation and impaired oxidative phosphorylation in retinal endothelial cells [46], although mitochondrial fragmentation alone is insufficient to universally predict dysfunctional metabolism. The observation of diabetes-induced mitochondrial fragmentation was rationalized by the critical finding of functional changes to oxidative phosphorylation (Figure 3, Figure 4 and Figure 6). Citrate synthase is a marker of mitochondrial content and its activity parallels electron transport chain capacity of the cell [37]. Steady-state mitochondrial content, however, is controlled by the relative flux of mitochondrial biogenesis and mitophagy which are, in turn, related to mitochondrial fission and fusion dynamics. This quality-control mechanism is useful to clear bioenergetically dysfunctional mitochondria by fission and subsequent mitophagy. It ensures a steady-state population of robust mitochondria capable of sustaining ATP synthesis rates over a wide range of metabolic demands. Indeed, diabetes-induced increases in mitophagy have been described in RPE cells with increased mitophagic flux attributed to ROS-dependent mitochondrial damage [47,48]. Our observations of diabetes-induced oxidative phosphorylation dysfunction (Figure 4), likely followed mitochondrial fragmentation (Figure 3), rationalize earlier reports of the increased mitophagic flux. Our data showed that diabetes-induced ASM upregulation led to an accumulation of ceramide in mitochondrial membranes that limits the maximal metabolic capacity of the respiratory chain. Combined with excessive electron supply from glucose and adequate oxygenation in the hyperglycemic stage of the diabetes, such a restriction stimulates ROS production. This metabolic insult then leads to the production of dysfunctional mitochondria, which stimulates the mitophagy pathway and, under continuously elevated ASM levels, results in a steady population of fragmented and bioenergetically impaired mitochondria in RPE cells.

Our current finding that desipramine treatment can rescue functional (RCR, Figure 4) and morphological (content, Figure 6) mitochondrial changes in diabetic-derived RPE cells demonstrates that ASM-dependent ceramide metabolism plays central role in diabetes-induced mitochondrial damage. Desipramine belongs to a class of antidepressants known as tricyclic amines which are functional inhibitors of ASM activity [31,32]. Although there are reports that at a high dose, desipramine can also interact with mitochondrial proteins directly, leading to impaired NADH oxidation, electron transport, and ATP synthase activity [49], these effects were not observed at the low dosage (15 μM) and short treatment time (1 h daily) use in this study. Indeed, our results showed no changes of the basal rate (Figure 4 and Appendix A) and substantial enhancement of the maximal oxidative phosphorylation function, which were not consistent with the direct effects of desipramine on mitochondrial oxidative phosphorylation machinery. The observed changes rather support the effect of desipramine via inhibition of ASM activity, leading to the depletion of mitochondrial ceramide, reversing its inhibitory effect on the oxidative phosphorylation and increasing RCR, as described here. 

As a gross measure of mitochondrial fitness, the whole-cell RCR is sensitive to a range of metabolic processes including substrate delivery, maximal electron transport chain capacity, proton leakage, and outer mitochondrial membrane integrity [30,50]. A greater than two-fold decrease in the whole-cell RCR of diabetic-derived RPE cells suggests substantial impairment of oxidative phosphorylation with a concomitant decrease in mitochondrial ATP-synthesis capacity and increase in mitochondrial ROS generation. It is remarkable that the mitochondrial functional impairment was retained despite culturing the cells for several generations under standard, euglycemic conditions. Whether the diabetes-induced RCR changes arise from direct inhibition of substrate delivery, electron transport, or the phosphorylation system, such as direct ceramide inhibition of Complex III or ceramide-mediated formation of outer mitochondrial membrane pores [18,19,20,21], is the subject of ongoing research. The sensitivity of diabetic-derived RPE cells to desipramine strongly suggests ASM-mediated ceramide inhibition of oxidative phosphorylation. Depletion of mitochondrial ceramide upon ASM inhibition, therefore, would reverse these effects, reduce oxidative stress, and favor retention of a robust mitochondrial population, as we saw in this work (Figure 4 and Figure 6).

## 4. Materials and Methods

### 4.1. Rodents

All animal procedures complied with the National Institutes of Health (NIH) Guide for the Care and Use of Laboratory Animals. Procedures received prior approval by IACUC at Michigan State University, approval #Busik08/17-151-00, 28 August 2017.

Diabetes was induced in male Sprague–Dawley rats (237–283 g) with a single intraperitoneal injection of streptozotocin (STZ) (65 mg/kg) (Sigma Aldrich, St. Louis, MO, USA) dissolved in 100 mM citric acid (pH = 4.5) (29). Body weights and blood glucose were monitored biweekly. Blood glucose concentration was maintained in the 20 mM range. Rats were used 7 weeks after diabetes induction. C57BL/6J ASM-deficient (ASM^−/−^) male mice and littermate wild-type controls at 6–8 weeks of age were used in the study.

### 4.2. Cell Culture

Primary human RPE were isolated according to standard procedures and cellular phenotype was confirmed by staining for ZO-1 and RPE65 markers [51]. ARPE-19 (ATCC CRL-2302) cells were grown in Dulbecco’s modified Eagle’s medium/F12 (1:1, *v*/*v*) supplemented with 10% fetal bovine serum and 1% penicillin/streptomycin at 37 °C in 95% relative humidity and 5% CO_2_. Primary human RPE cells were used at Passages 4–8.

### 4.3. Mitochondrial Isolation 

Mitochondria were isolated according to previously described protocols with minor modifications [36,52]. Briefly, cells were resuspended in ice-cold mitochondrial isolation buffer (mIB) and lysed for 20 s with a Scilogex D160 homogenizer (Scilogex, Rocky Hill, CT, USA) equipped with a 5 mm diameter probe operated at 18,000 rpm. The homogenate was brought to 30 mL with fresh mIB and centrifuged at 1000× *g* for 10 min at 4 °C. The supernatant was reserved, and the pellet was homogenized and centrifuged as above. The pooled supernatants were centrifuged at 8000× *g* for 15 min, and the mitochondrial pellet was washed with fresh mIB and subjected to further processing as indicated. Where required, half of the isolated mitochondrial sample was further purified via sucrose step-density gradient ultracentrifugation without modifications using a Sorvall M120 SE Micro-Ultracentrifuge (S55S-1155, ThermoFisher Scientific, Waltham, MA, USA) [52].

### 4.4. Mass Spectrometry

Mitochondria on dry ice were subjected to lipid extraction with chloroform, methanol, and water as previously described [26]. Dried lipid extracts were washed with 10 mM ammonium bicarbonate solution to remove salts and buffer contaminants, and then dried under a vacuum and resuspended in methanol by normalizing volumes to total mitochondrial protein. Immediately before analysis, mitochondrial lipids were diluted 5-fold by drying aliquots in a speed-vac centrifuge and resuspending in five volumes of isopropanol/methanol/chloroform (4:2:1, *v*:*v*:*v*) containing 20 mM ammonium formate. Lipids were analyzed by high-resolution/accurate mass spectrometry and tandem mass spectrometry in positive- and negative-ionization modes on an LTQ-Orbitrap Velos mass spectrometer (Thermo Scientific, Waltham, MA, USA) [26]. A TriVersa Nanomate (Advion, Ithaca, NY, USA) functioned as a nano-electrospray ionization source and autosampler. The nESI spray voltage was held at 2.4 kV and nESI gas pressure was 0.3 psi. The 96 well sample plate (Eppendorf, Hamburg, Germany) was held at 12 °C. Sphingolipid species were quantified as their formate adducts in negative-ionization mode against spiked synthetic sphingolipid internal standards of Cer(30:1) and SM(30:1) (Avanti Polar Lipids, Alabaster, AL, USA) at 250 femtomole/microliter [53]. Sphingolipid structures were confirmed by higher-energy collisional dissociation MS/MS in positive ionization mode. Each mass spectrum was subjected to offline mass recalibration using Thermo Xcalibur software to correct for any instrumental drift in mass calibration. Lipid peaks were subjected to isotope correction, identified, and quantified against sphingolipid internal standards using LIMSA software [54] as previously described [26].

### 4.5. Immunocytochemistry and Mitochondrial Morphology

Cells were washed three times with PBS and fixed for 15 min at room temperature with Histochoice fixative (Sigma, cat no. H2904). Cells were permeabilized with 0.1% Triton X-100 in PBS for 20 min and blocked with 1.5% BSA, 1% Tween-10 in PBS (PBST) overnight at 4 °C. Blocked samples were incubated with anti-ceramide antibody (Sigma, cat no. 8104) at a 1:100 dilution at 4 °C overnight. After three washes with PBST, cells were incubated with anti-mouse secondary antibody conjugated to Alexafluor 488 at a 1:100 dilution. Cells were counterstained with DAPI and imaged on a Nikon Eclipse TE2000 (Nikon Instruments Inc., Melville, NY, USA) equipped with a Photometrics CoolSNAP HQ2 camera (Photometrics, Tucson, AZ, USA). Fluorescence intensity was quantified with ImageJ software (version 1.53a, National Institutes of Health, Bethesda, MD, USA). For mitochondrial morphology determination, primary human RPE were grown on coverslips and stained with 50 nM MitoTracker Green at 37 °C for 30 min. After washing with PBS, cells were imaged on a Ziess LSM880 microscope (Zeiss, Oberkochen, Germany). Mitochondrial length, as a marker of fragmentation, was determined by measuring the major axis of individual mitochondria in a 5 × 5 μm square from a randomly selected cell in the field of view. Five cells from three fields of view were selected from each sample to represent the cellular population. 3D animations of the z-stacked images were created use the 3D projection command in ImageJ, setting layer height to 0.17 μm and a full 360° rotation (Appendix A).

### 4.6. Quantitative Real-Time Polymerase Chain Reaction 

Total cellular RNA extraction and RT-PCR were performed as previously described [15]. Human gene-specific primers for acid sphingomyelinase, interleukin *1β* (*IL-1β*), interleukin 6 (*IL-6*), intercellular adhesion molecule (ICAM1), and vascular endothelial growth factor (VEGF) were used to measure gene expression. Results were normalized to cyclophilin A.

### 4.7. Western Blot Analysis

Protein extraction and Western blots were carried out using the NuPAGE system as previously described [3]. Fractions of mitochondrial isolates were normalized by suspension volume and quantitated relative to voltage-dependent anion channel (VDAC) intensity. Primary antibodies against LAMP-1 (SC-20011, Santa Cruz, Dallas, TX, USA), VDAC (PAI-954A, Invitrogen, Waltham, MA, USA), and ASM were used at 1:1000 dilution. Anti-ASM antibody was a generous gift from Richard Kolesnick. Secondary antibodies against rabbit IgG (926-68073, Odyssey, Lincoln, NE, USA) and mouse IgG (610-731-124, Rockland Immunochemicals, Limerick, PA, USA) were used at 1:10,000 dilution. Bands were imaged on a LiCor Odyssey imaging system. Densitometric analysis was performed in ImageJ software after splitting the RGB image into individual channels and a background subtraction using a rolling ball radius of 16.3 pixels.

### 4.8. Citrate Synthase Activity

Enzymatic activity was measured using a citrate synthase activity assay kit (Sigma, cat no. CS0720) per the manufacturer’s instructions using a 96 well plate. Citrate synthase activity was normalized to total protein content, measured using the Bradford assay (BioRad, Hercules, CA, USA).

### 4.9. Microrespirometry

Cellular respirometry was measured as previously described [28]. Briefly, cells were seeded on-chip at a density of 860 cells/mm^2^ and cultured overnight under standard cell culture conditions at 37 °C, 95% relative humidity, 5% CO_2_. Microrespirometer chips were assembled immediately before the measurement and the cells were perfused with the basal respiration buffer, which consisted of DPBS with calcium and magnesium supplemented with 10 mM glucose, 10 mM lactate, 1 mM pyruvate, and 0.2% bovine serum albumin. Respiratory control ratio [30] was determined from sequential respiration measurements in the leak buffer, consisted of respiration buffer supplemented with 2.5 μM oligomycin, and the uncoupling buffer, consisted of leak buffer supplemented with 5 μM carbonyl cyanide m-chlorophenylhydrazone (CCCP). Measurements in inhibition buffer, consisting of uncoupling buffer supplemented with 5 mM potassium cyanide, were used for correction of non-respiratory oxygen consumption. Perfusion was controlled using a syringe infusion pump (KD Scientific, Holliston, MA, USA) operating at a flow rate of 10 μL/min at room temperature. Activity determinations were performed under stationary buffer conditions for 5–10 min, maintaining oxygen concentrations above 150 μM. 

## Figures and Tables

**Figure 1 ijms-21-03830-f001:**
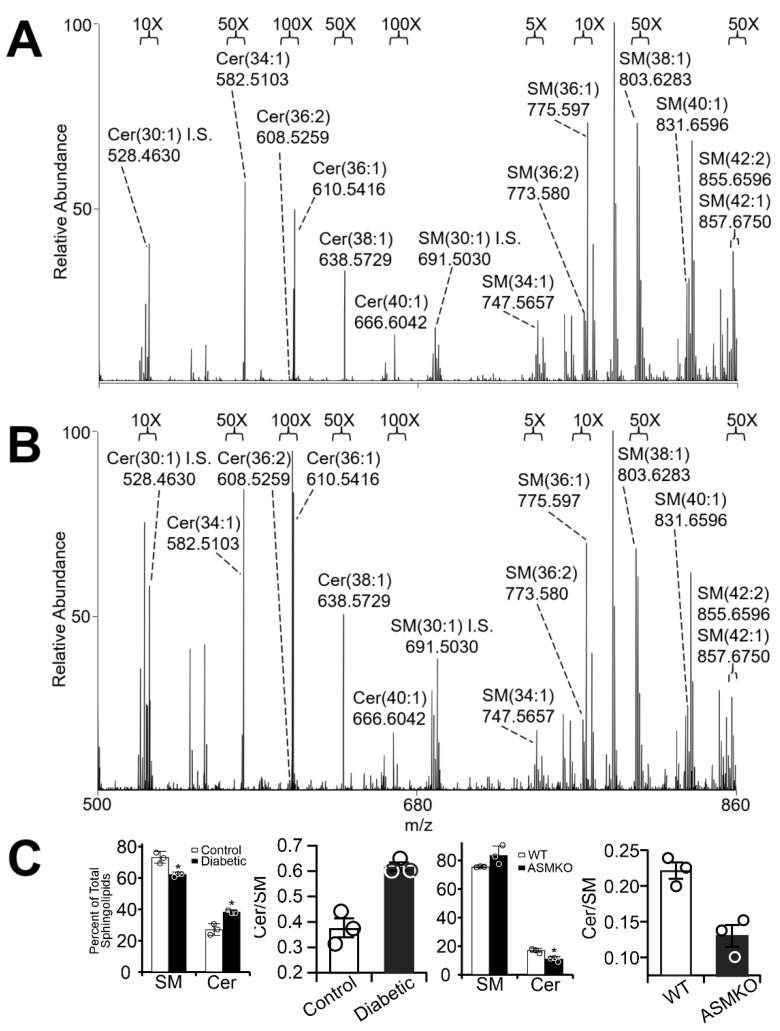
Negative-ion high-resolution/accurate mass spectrometric quantification of sphingolipids in retinal mitochondria. Mass spectra of control (**A**) and diabetic (**B**) rat retinal mitochondrial sphingolipids after 7 weeks of diabetes. Full-scan MS spectra are shown from mitochondrial lipids analyzed by negative-ionization mode direct-infusion nano-ESI mass spectrometry. Sphingolipids are shown under magnifications indicated at the top of each panel. Abundant non-labeled peaks correspond to phospholipids. “I.S.” indicates internal standards. (**C**) Quantification of total sphingomyelin (SM), ceramide (Cer), and the Cer/SM ratio based on mass spectrometry analysis of mitochondria from control and diabetic rat retinas (**left** panel), and wild type (WT) and acid sphingomyelinase knock out (ASMKO) mouse retinas (**right** panel). * *p* < 0.05, *n* = 3.

**Figure 2 ijms-21-03830-f002:**
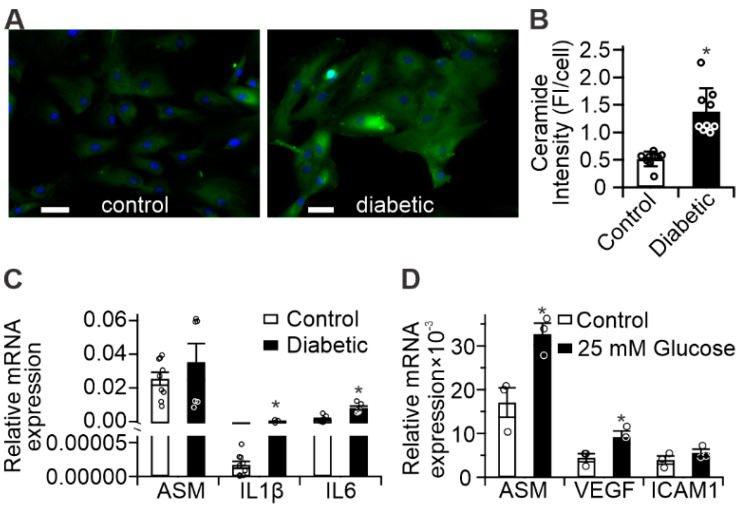
Diabetes-induced pro-inflammatory changes in human RPE (**A**) Representative images of control- and diabetic-derived retinal pigment epithelial (RPE) cells showing ceramide (green) and nuclear staining (blue). Scale bars = 50 µm; (**B**) Quantitation of ceramide-staining fluorescence intensity from panel (A). *n* = 9, error bars = S.D., * *p* < 0.05; (**C**) Inflammatory gene expression in diabetic-derived RPE (black bars) compared to control (white bars); (**D**) Upregulation of inflammatory gene expression in control RPE treated with 25 mM glucose for 72 h (black bars) compared to untreated cells (white bars). * *p* < 0.01, *n* = 6.

**Figure 3 ijms-21-03830-f003:**
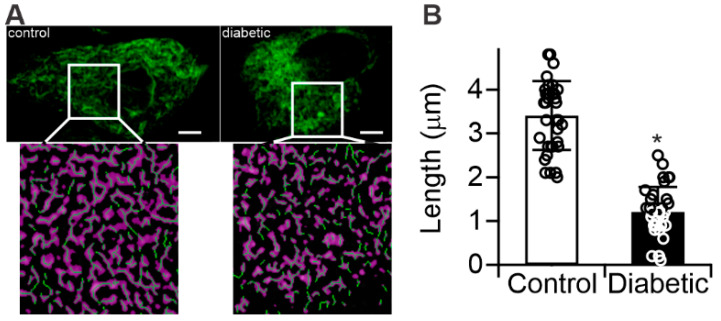
Structural analysis of human RPE mitochondria. (**A**) Mitochondrial morphology determined by MitoTracker Green staining of control- and diabetic-derived RPE. Inset = skeletonized (green lines) binary mask (purple) of deconvoluted photomicrographs highlighting mitochondrial morphology. Scale bars = 5 µm; (**B**) Quantitation of average mitochondrial length. *n* = 3, * *p* < 0.05.

**Figure 4 ijms-21-03830-f004:**
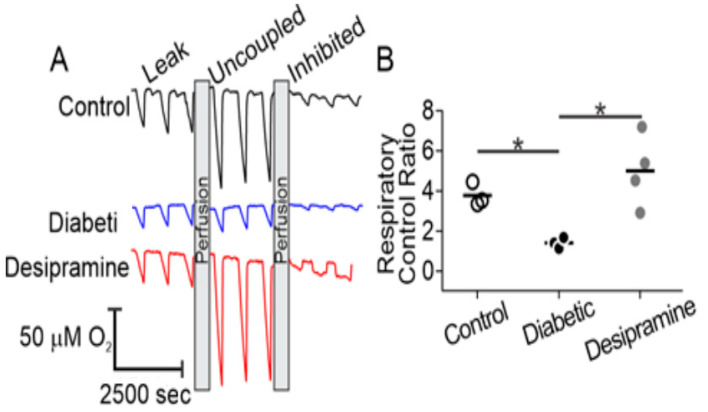
Microrespirometric analysis of human RPE cells. (**A**) Representative oxygen concentration traces of control (black), diabetic (blue), and desipramine-pretreated diabetic (red) RPE cells. Respirometry was performed in the presence of oligomycin (leak), carbonyl cyanide m-chlorophenylhydrazone (CCCP, uncoupled), and potassium cyanide (KCN, inhibited). See text for details; (**B**) Respiratory control ratios of control (white circles), diabetic (black circles), and desipramine-pretreated diabetic groups (gray circles). * *p* < 0.05, *n* = 3–4.

**Figure 5 ijms-21-03830-f005:**
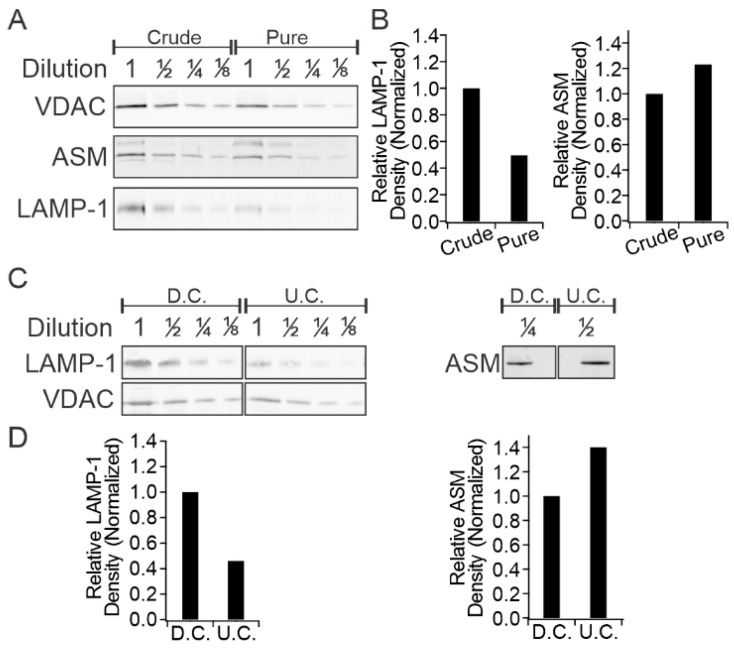
Colocalization between ASM and mitochondrial markers. (**A**) Western blot analysis of human RPE cell mitochondria at varying levels of purity and (**B**) associated optical density quantitation; (**C**) Western blot analysis of ARPE-19 cell mitochondria at varying purity levels and associated optical density quantitation (**D**). LAMP-1 = lysosome associated membrane protein 1, VDAC = voltage dependent anion channel, crude = one 8000× *g* centrifugation, pure = three 8000× *g* centrifugations, D.C. = differential centrifugation, U.C. = sucrose step-density ultracentrifugation.

**Figure 6 ijms-21-03830-f006:**
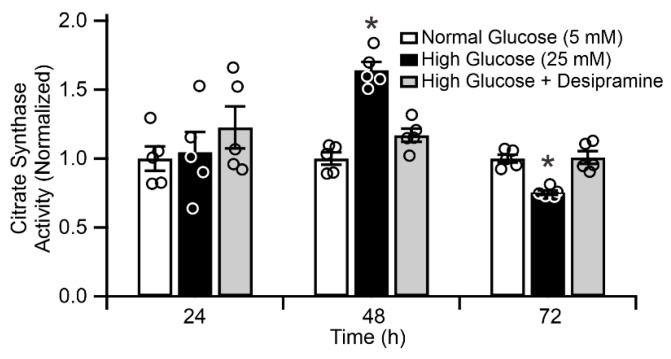
Citrate synthase activity in ARPE-19 cells. White bars = control cells, black bars = 25 mM glucose-treated cells, gray bars = 25 mM glucose-treated cells with daily 1 h treatment with 15 µM desipramine. * *p* < 0.05, *n* = 5.

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
