# Peer review of "Mitochondrial Ceramide Effects on the Retinal Pigment Epithelium in Diabetes"

_ijms, 2020, doi:10.3390/ijms21113830_

Round 1
Reviewer 1 Report
The study of Yan Letvisky and his colleagues untitled « Mitochondrial Ceramides Effects on the Retinal Pigment epithelium in Diabetes” describes how diabetes causes a metabolic deficit in the retinal pigmentary epithelium (RPE) by affecting the sphingolipids, specifically by increasing the level of mitochondrial ceramide, causing mitochondrial dysfunction. The latter is likely to induce oxidative stress leading to inflammation and ultimately death of the RPE. The authors show that ASM is a key enzyme in this process in response to ceramide-producing hyperglycemia. The whole work is original, well done and well transcribed. Nevertheless, the manuscript can be improved by a few minor modifications.
- Figue 1 shows the changes in ceramide and sphingomyelin in diabetic rat retina and in ASM knockout mouse retina, confirming the role of ASM in mitochondrial sphingolipid metabolism. Because sphingolipid and ceramide metabolites are multiples, showing their MS profiles could help to assess the relevance of this quantification.
- Figure 2 examines the ceramide content of RPE cells. Surprisingly, the distribution is not homogeneous in diabetic condition, despite the almost 3 fold increase, which is higher than the increase of ceramide in mitochondria from diabetic rat. Is this heterogeneity a failure of labelling? Do you have an accurate interpretation of this result?
3 The mitochondrial morphology is poorly visible on the photographs. The quantification using ImageJ further supports the observation.
- ASM expression was described in the plasma membrane and lysosomes. How a plasma membrane biomarker is distributed within the two-steps purification?
- The accumulation of mitochondrial ceramides appears to be contradictory with the previously published decrease due to its glycosylation (). Moreover, despite the culture of RPE cells in normoglycemia, a response of ceramide expression in diabetic RPE cells compared to non-diabetic cells is observed. Nevertheless, high glucose supplemented RPE cells continue to respond by ASM and ceramide accumulation. These data may suggest cell culture-dependent metabolic regulation. It would be interesting to test the glucose sensitivity of EPR primary cultures with no passage and reduced culture time.
Author Response
To: Radu Danila
Assistant Editor
International Journal of Molecular Sciences
Re: ijsm-804330 minor revisions
Dear Dr. Danila,
Our manuscript titled “Mitochondrial Ceramide Effects on the Retinal Pigment Epithelium in Diabetes” was recently evaluated by IJMS. The reviewers found it to be “original, well done and well transcribed” and commented that “the study was well designed, and data were acquired using their well-established biotechnologies”, however the reviewers felt that the manuscript could be further improved with minor revisions. The revised manuscript has been generally edited for clarity and to address the reviewer comments. Additionally, methods were re-written, and all figures have been modified to include larger font sizes.
We thank the reviewers for their constructive criticism, and we believe we were able to address all the reviewer’s comments as follows:
- Figure 1 shows the changes in ceramide and sphingomyelin … Because sphingolipid and ceramide metabolites are multiples, showing their MS profiles could help to assess the relevance of this quantification.
Thank you for this suggestion, to address this comment, we have included high resolution/accurate mass spectra of mitochondrial sphingolipids from control and diabetic rat retinas in Figure 1.
- Figure 2 examines the ceramide content of RPE cells. Surprisingly, the distribution is not homogeneous in diabetic condition, despite the almost 3 fold increase, which is higher than the increase of ceramide in mitochondria from diabetic rat. Is this heterogeneity a failure of labelling? Do you have an accurate interpretation of this result?
Thank you for this astute comment. The whole cell ceramide levels and distribution are, indeed, different from mitochondrial ceramide. As we have previously demonstrated(1-3), there are ASM-dependent ceramide pools in the plasma membrane, endoplasmic reticulum, lysosomes and, as presented in this study, mitochondria. Thus, the increase in total cellular ceramide was used as an indicator of overall activation of ASM, however mitochondrial preparations were used to determine the specific contribution of mitochondrial ceramide to total cellular ceramide levels. Moreover, the RPE cells are not a part of the whole retina preparation. We further strengthened the point in the manuscript that RPE cell are left behind during retinal isolation. The major component of the whole retina mitochondrial preparations are photoreceptor mitochondria. Moreover, diabetes-induced increase in ASM expression and activity is the highest in the cells comprising BRB, namely REC and RPE cells(1-3), with smaller changes in the Muller cells and microglia and no observed changes in the photoreceptors(3). The effect in the whole retina mitochondria preparations is thus diluted by the large contribution from non-changing photoreceptor mitochondria with a smaller contribution of the increased mitochondrial ceramide from endothelial, Muller and microglia cells. We have added a clarifying statement to the manuscript.
- The mitochondrial morphology is poorly visible on the photographs. The quantification using ImageJ further supports the observation.
- In Figure 3, the mitochondrial length can NOT be observed by using a low-resolution picture nor low-power microscopy. The 5um-scale bar is not sufficient.
Thank you for these comments. We agree that the figure left ambiguity in interpretation. We have updated the figure to include larger insets which now show the highlighted regions as skeletonized binary masks of the deconvoluted photomicrographs. As the main goal of the manuscript is to describe the functional changes to retinal cells in diabetes, and as diabetes-induced mitochondrial fragmentation is a well-documented morphological change, we chose to use fluorescence confocal microscopy to estimate the changes as opposed to the labor-intensive and technically challenging methods relying on 3D electron microscopy reconstruction.
- In Figure 1, lack of description of detailed materials and methods. How many grams of tissue samples? How many micrograms of mitochondrial proteins? What was the signal intensity of each ceramide? How the calculate total ceramide concentration?
We apologize for the lack of appropriate Methods description; we have updated the manuscript to include more experimental details.
- In Figure 4A, improve the data interpretation of the micro-respirometric analysis.
We appreciate this comment, the details are now included in the manuscript.
- Please discuss about the missing link between lysosomal ASM and mitochondrial ceramide. Why not nSMase, why not ceramide synthase?
Thank you for this comment. Indeed, several sphingolipid enzymes, such as nSMase, ceramide synthase and ceramidase, have been shown to induce mitochondrial ceramide accumulation in other systems. ASM was studied here owing to its pivotal role in the BRB breakdown characteristic of DR as well as its central role in stress-mediated ceramide generation. Indeed, we have previously demonstrated in the animal and cell culture models of DR that it is the ASM, rather than nSMase that is increased in the retina and retinal cells(3). Moreover, we have shown that there was no effect of de novo ceramide production pathway inhibition (ceramide synthase inhibition using Fumonisin B1) on cytokine-induced pro-inflammatory changes in the retina and retinal cells(2). We have added clarifying statements to the discussion.
- ASM expression was described in the plasma membrane and lysosomes. How a plasma membrane biomarker is distributed within the two-steps purification?
We appreciate the opportunity to clarify this important question. The mitochondrial isolation method that is used in this study provides highly purified mitochondria devoid of the plasma membrane contamination. As the lysosomes and mitochondria have very similar size, shape, and density characteristics, they are much harder to separate and thus we paid special attention to the potential lysosomal contamination in this study. This point is strengthened in the manuscript.
- The accumulation of mitochondrial ceramides appears to be contradictory with the previously published decrease due to its glycosylation.
Thank you for pointing this out. The increased glycosylation in the diabetic retina was shown to be due the increase in uridine diphosphate glucose (UDP-glucose) production through the pentose pathway, rather than the changes in enzymatic activity. As pentose pathway occurs in the cytoplasm, the increase in glucosylceramide production due to higher UDP-glucose availability would be limited to the cytoplasm, rather than the mitochondria.
- Moreover, despite the culture of RPE cells in normoglycemia, a response of ceramide expression in diabetic RPE cells compared to non-diabetic cells is observed. Nevertheless, high glucose supplemented RPE cells continue to respond by ASM and ceramide accumulation. These data may suggest cell culture-dependent metabolic regulation. It would be interesting to test the glucose sensitivity of RPE primary cultures with no passage and reduced culture time.
We apologize for not including a clear explanation of the metabolic memory or legacy phenomenon observed here. This phenomenon was first described in diabetic patients as prolonged effect of prior glucose control levels on the development of diabetic complications, even after the new glucose control levels are established(4). The metabolic memory is attributed to epigenetic changes and it is well accepted in the diabetic complications field(4-7). It was shown to occur in the animal models, as well as in cell culture models, where the cells isolated from diabetic donor retina or animal model keep diabetic “metabolic memory” for several passages(4-9). Human control and diabetic donor cells were previously shown to display metabolic memory characteristics right after the isolation (passage 1) and for up to 8 passages(4). Passages 4-8 were used in this study because the initial cell expansion was required to obtain enough cells to perform the experiments.
We have now included a more detailed explanation of this phenomenon in the manuscript.
- Maria Tikhonenko, Todd A. Lydic, Madalina Opreanu, Sergio Li Calzi, Svetlana Bozack, Kelly M. McSorley, Andrew L. Sochacki, Matthew S. Faber, Sugata Hazra, Shane Duclos, Dennis Guberski, Gavin E. Reid, Maria B. Grant, Busik JV. N-3 Polyunsaturated Fatty Acids Prevent Diabetic Retinopathy by Inhibition of Retinal Vascular Damage and Enhanced Endothelial Progenitor Cell Reparative Function. PLOS ONe. 2013;8(1):e55177. doi: 10.1371/journal.pone.0055177.
- Opreanu M, Lydic TA, Reid GE, McSorley KM, Esselman WJ, Busik JV. Inhibition of cytokine signaling in human retinal endothelial cells through downregulation of sphingomyelinases by docosahexaenoic acid. Invest Ophthalmol Vis Sci. 2010;51(6):3253-63. Epub 2010/01/15. doi: 10.1167/iovs.09-4731. PubMed PMID: 20071681; PMCID: 2891477.
- Opreanu M, Tikhonenko M, Bozack S, Lydic TA, Reid GE, McSorley KM, Sochacki A, Perez GI, Esselman WJ, Kern T, Kolesnick R, Grant MB, Busik JV. The unconventional role of acid sphingomyelinase in regulation of retinal microangiopathy in diabetic human and animal models. Diabetes. 2011;60(9):2370-8. Epub 2011/07/21. doi: 10.2337/db10-0550. PubMed PMID: 21771974; PMCID: 3161322.
- Kowluru RA. Diabetic retinopathy, metabolic memory and epigenetic modifications. Vision Res. 2017;139:30-8. Epub 2017/07/13. doi: 10.1016/j.visres.2017.02.011. PubMed PMID: 28700951.
- Alivand MR, Soheili ZS, Pornour M, Solali S, Sabouni F. Novel Epigenetic Controlling of Hypoxia Pathway Related to Overexpression and Promoter Hypomethylation of TET1 and TET2 in RPE Cells. J Cell Biochem. 2017;118(10):3193-204. Epub 2017/03/03. doi: 10.1002/jcb.25965. PubMed PMID: 28252217.
- Desjardins D, Liu Y, Crosson CE, Ablonczy Z. Histone Deacetylase Inhibition Restores Retinal Pigment Epithelium Function in Hyperglycemia. PLoS One. 2016;11(9):e0162596. Epub 2016/09/13. doi: 10.1371/journal.pone.0162596. PubMed PMID: 27617745; PMCID: PMC5019386.
- Dolinko AH, Chwa M, Atilano SR, Kenney MC. African and Asian Mitochondrial DNA Haplogroups Confer Resistance Against Diabetic Stresses on Retinal Pigment Epithelial Cybrid Cells In Vitro. Mol Neurobiol. 2020;57(3):1636-55. Epub 2019/12/08. doi: 10.1007/s12035-019-01834-z. PubMed PMID: 31811564; PMCID: PMC7123578.
- Peng QH, Tong P, Gu LM, Li WJ. Astragalus polysaccharide attenuates metabolic memory-triggered ER stress and apoptosis via regulation of miR-204/SIRT1 axis in retinal pigment epithelial cells. Biosci Rep. 2020;40(1). Epub 2020/01/03. doi: 10.1042/BSR20192121. PubMed PMID: 31894851; PMCID: PMC6974424.
- Roy S, Sala R, Cagliero E, Lorenzi M. Overexpression of fibronectin induced by diabetes or high glucose: phenomenon with a memory. Proc Natl Acad Sci U S A. 1990;87(1):404-8. Epub 1990/01/01. doi: 10.1073/pnas.87.1.404. PubMed PMID: 2296596; PMCID: PMC53272.
Reviewer 2 Report
In this study, Levitsky et al. showed that ceramide accumulation in mitochondria isolated from Streptozotocin (STZ)-induced diabetic rat retinas. Later the authors focused on human retinal pigment epithelium (RPE) cells. Diabetes-induced proinflammatory cytokines were associated with mitochondrial fission and dysfunction. Treatment with desipramine, an inhibitor of acid sphingomyelin (ASM), the impairment of metabolic functions were rescued. These findings support that ceramide overproduction may be an essential issue in diabetes; treatment targeting on ASM may protect diabetic retinopathy. Overall, the study was well designed, and data were acquired by using their well-established biotechnologies. The only concern of this paper is the quality of the presentation.
- In Figure 3, the mitochondrial length can NOT be observed by using a low-resolution picture nor low-power microscopy. The 5um-scale bar is not sufficient.
- In Figure 1, lack of description of detailed materials and methods. How many grams of tissue samples? How many micrograms of mitochondrial proteins? What was the signal intensity of each ceramide? How the calculate total ceramide concentration?
- In Figure 4A, improve the data interpretation of the micro-respirometric analysis.
- Please discuss about the missing link between lysosomal ASM and mitochondrial ceramide. Why not nSMase, why not ceramide synthase?
- Some of the font size in the figures are too small.
Author Response

(The authors gave the same response as above.)
